# Atherogenic Index of Plasma (AIP) and Its Association with Fatty Liver in Obese Adolescents

**DOI:** 10.3390/children10040641

**Published:** 2023-03-29

**Authors:** Hüseyin Dağ, Fatih İncirkuş, Okan Dikker

**Affiliations:** 1Division of Pediatrics, Istanbul Prof. Dr. Cemil Taşcıoğlu City Hospital, University of Health Sciences, Istanbul 34384, Turkey; 2Division of Medical Biochemistry, Istanbul Prof. Dr. Cemil Taşcıoğlu City Hospital, University of Health Sciences, Istanbul 34384, Turkey

**Keywords:** atherogenic index of plasma, fatty liver, obesity, vitamin D

## Abstract

Background: The atherogenic index of plasma (AIP) is the base-10 logarithmic conversion of the triglyceride to high-density lipoprotein cholesterol ratio [AIP = log10 (triglyceride/HDL cholesterol)]. Some studies have found a link between low serum vitamin D levels, AIP, and fatty liver. This study was conducted to evaluate the relationship between AIP levels, fatty liver, and vitamin D levels in obese adolescents aged 10–17 years. Methods: This study included 136 adolescents, including 83 obese and 53 healthy controls, in the age range of 10–17 years. Thirty-nine of the obese adolescents had fatty livers. Those with ultrasonography grades 2 or 3 of fat were in the fatty liver group. The AIP value was calculated as the logarithmic conversion of the ratio (triglyceride/HDL cholesterol) at the base of 10. Vitamin D and other laboratory tests were analyzed biochemically. Statistical evaluations were made with the SPSS program. Results: The AIP, body mass index (BMI), homeostatic model assessment for insulin resistance (HOMA-IR), and insulin averages of obese adolescents with fatty liver were significantly higher than those of obese adolescents without fatty liver and the healthy control group (*p* < 0.05). Again, the mean AIP of obese patients without fatty liver was pointedly higher than that of the healthy control group (*p* < 0.05). There was a positive, moderate relationship between AIP and BMI, AIP and HOMA-IR, and AIP and insulin levels (*p* < 0.05), whereas there was a negative, moderate (37.3%) relationship between AIP and vitamin D (*p* = 0.019). Conclusion: AIP levels were higher in obese adolescents, and this increase was higher in obese adolescents with fatty liver in this study. Moreover, we detected a negative correlation between AIP and vitamin D levels and a positive correlation with BMI, insulin resistance, and insulin levels. Based on our data, we concluded that AIP can be a useful predictor of fatty liver in obese adolescents.

## 1. Introduction

Non-alcoholic fatty liver disease (NAFLD) is a major and common chronic liver disease in children. Studies on obese children have reported that the NAFLD prevalence varies up to 85% [1]. The prevalence of chronic liver disease in adolescents and youthful adults has more than doubled in the last thirty years, mostly due to an increase in the prevalence of NAFLD [2]. In recent years, there has been a dramatic rise in the prevalence of NAFLD with increasing obesity rates [3,4]. NAFLD is also associated with type 2 diabetes mellitus, hyperlipidemia, and insulin resistance. The disease may progress to fibrosis, cirrhosis, and liver carcinoma in the liver [5]. In the pathogenesis of NAFLD, excessive triglyceride accumulation in the liver is a prerequisite, and there is an imbalance between lipid intake and lipid excretion that eventually leads to oxidative stress and hepatocyte damage [6].

NAFLD is associated with a high cardiovascular disease (CVD) risk and an atherogenic lipid profile, including increased triglycerides, LDL cholesterol, and very low-density lipoprotein cholesterol (VLDL cholesterol) levels and lower HDL cholesterol levels [7,8,9]. CVD is a common cause of death in NAFLD patients [10]. NAFLD is a self-sufficient risk factor for atherosclerotic CVD development, and, by the same token, patients with NAFLD have chronic inflammation and increased oxidative stress levels, indicating CVD risk [11]. Inflammation, irregularity of lipoprotein metabolism, and oxidative stress lead to the collection of cholesterol esters in the macrophages of the arterial wall in the pathogenesis of atherosclerosis [12].

Semiquantitative grading system ultrasonography (USG) is an acceptable tool for NAFLD screening. USG can detect and rate NAFLD. It is non-invasive, fast, cost-effective, and easily accessible. However, USG has moderate accuracy in detecting NAFLD [13]. However, total cholesterol/HDL cholesterol, LDL cholesterol/HDL cholesterol, and triglyceride/HDL cholesterol rates are preferable indicators of metabolic and clinical links among lipid fractions and have better predictive powers than parameters used independently of disease conditions [14]. Researchers have examined the relationship between another lipid ratio, the atherogenic index of plasma (AIP) and NAFLD, and they found a relationship between them [15,16,17,18,19,20]. Studies show that AIP is a strong indicator of NAFLD and can predict CVD development [15,16,17,18,19]. However, studies in the pediatric age group are limited. There is an important positive correlation between AIP and steatosis grade; AIP may be a risk indicator for NAFLD and CVD in obese children [20].

The AIP was defined by Dobiasova and Frohlich in 2001 as a sensitive and powerful index reflecting the interaction between atherogenic and non-atherogenic lipoproteins. The AIP is the base-10 logarithmic conversion of the triglyceride to high-density lipoprotein–cholesterol ratio [AIP = log_10_ (triglyceride/HDL cholesterol)] [21]. The AIP is a powerful indicator for estimating the risk of atherosclerosis and cardiovascular disease [22,23,24]. Compared to the routine lipid profile, the AIP is a better predictor of atherosclerosis than LDL cholesterol [21]. The AIP prevents inconsistent evaluation of different lipid components and simplifies the routine estimation task [25].

Many studies showed the connection between AIP and vitamin D levels [26]. Data in the literature suggest that decreasing serum vitamin D levels can lead to fatty liver disease. Hypovitaminosis D was associated with the severity and ratio of NAFLD [27]. Based on the literature data, for the first time, we conducted this study to examine the relationship between AIP values and NAFLD and vitamin D levels in obese adolescents in the 10–17 adolescent age group.

## 2. Materials and Methods

### 2.1. Patients and Design

This study included 136 adolescents, 83 obese and 53 healthy controls, aged 10–17, who were admitted to the pediatrics division of Dr. Cemil Taşcıoğlu at City Hospital in İstanbul between 15 November 2022 and 15 January 2023. Adolescents with a body mass index (BMI) of 95th percentile and above according to age and sex were ‘obese’. Thirty-nine of the obese adolescents had fatty livers. Those with grade 2 or 3 fatty liver according to USG were in the NAFLD-fatty liver group. Those with grade 1 fatty liver were not included in the study. The healthy control group consisted of 53 patients without fatty liver according to USG whose BMI was in the normal percentile for their age and sex. The AIP value was the base-10 logarithmic conversion of the ratio (triglyceride/HDL cholesterol). Vitamin D and other laboratory tests were analyzed biochemically. All laboratory tests, sex, age, and BMI levels were statistically checked among the groups. Correlations were examined between the obese adolescent group and the group with liver fat. Smoking, obesity, the presence of chronic disease other than fatty liver, infection, those receiving any medication, and those receiving vitamin D supplements within the last six months were not in the study.

We performed a power analysis before beginning this study. As a result of the power analysis using the G*Power program, when the effect size is d: 1.010 and the standard deviation is 0.27 for AIP, the sample number for power = 0.80 and α = 0.05 was determined for a minimum of 17 individuals for each group.

### 2.2. Calculation of the Parameters

AIP = log_10_ (triglyceride/HDL cholesterol) [21].

Homeostatic model assessment of insulin resistance (HOMA-IR): for insulin resistance (insulin IU/L × glucose mg/dL)/405 [28].

LDL cholesterol level calculations were in line with Friedewald’s formulation [LDL cholesterol = total cholesterol (HDL cholesterol) – (triglyceride/5)].

### 2.3. Laboratory Tests

The colorimetric method measured glucose, urea, creatinine, aspartate aminotransferase (AST), alanine aminotransferase (ALT), C-reactive protein (CRP), total cholesterol, triglyceride, HDL cholesterol, LDL cholesterol, and calcium. The chemiluminescent immunoassay method evaluated thyroid stimulating hormone (TSH), free T4 (fT4), 25-hydroxyvitamin D3, and insulin in an autoanalyzer (Roche Brand, Cobas 8000 model). HbA1c was measured in an autoanalyzer (Bio-Rad, Variant II turbo) by high-performance liquid chromatography.

### 2.4. Statistical Investigation

The IBM SPSS Statistics 22 for statistical analysis (SPSS IBM, Turkey) program was utilized for statistical analyses. The compatibility of the parameters for the normal distribution was measured by the Kolmogorov–Smirnov and Shapiro–Wilks tests. In addition to descriptive statistical methods (mean, standard deviation, and frequency), we compared parameters with normative distribution between the two groups’ Student’s *t*-test and the quantitative data. The Mann–Whitney U-test compared the parameters that did not have a normative distribution between the two groups. A one-way ANOVA made intergroup comparisons of normally distributed parameters, and a Tukey’s HSD test determined the group causing the difference. The Kruskal–Wallis made intergroup comparisons of non-normally distributed parameters, and Dunn’s test determined the group causing the difference. Pearson’s correlation analysis scrutinized the relationships between parameters that were compatible with a normal distribution, whereas Spearman’s rho correlation analysis examined the links between parameters that do not have a normal distribution. The chi-square test compared qualitative data. The most appropriate cut-off point was chosen based on the receiver operating characteristic (ROC) curve analysis. The statistical significance was *p* < 0.05.

### 2.5. Ethical Approval

The research was acknowledged by Prof. Dr. Cemil Taşcıoğlu City Hospital Ethics Committee (14/11/2022-314).

## 3. Results

This study had 136 adolescent participants, including 62 (45.6%) girls and 74 (54.4%) boys, aged between 10 and 17 years. The mean age was 13.07 ± 1.94 years. Adolescents were in two groups: obese (n = 83) and control (n = 53). There was no statistically significant difference between the groups in mean age and sex distribution (*p* > 0.05). The mean BMI of obese adolescents was significantly higher than that of the control group (*p* < 0.05).

The AST, ALT, CRP, cholesterol, triglyceride, LDL cholesterol, HOMA-IR, insulin, cholesterol/HDL, triglyceride/HDL, AIP, cholesterol/triglyceride levels of obese adolescents were significantly higher than those of the healthy control group (*p* < 0.05). The HDL cholesterol levels of obese adolescents were considerably lower than those of the healthy control group (*p* < 0.05).

The biochemical parameters and Calculated indexes between the obese adolescent group and the healthy control group were presented in Table 1.

The AIP, cholesterol/HDL, triglyceride/HDL, and cholesterol/triglyceride mean of obese adolescents with fatty liver were significantly higher than those of obese people without fatty liver and the healthy control group (*p* < 0.05). Again, the average AIP, cholesterol/HDL cholesterol, and triglyceride/HDL cholesterol of obese patients without fatty liver were considerably higher than those of the healthy control group (*p* < 0.05) (Table 2).

The mean BMI, HbA1c, HOMA-IR, and insulin levels of those with fatty liver were considerably higher than those without fatty liver and the healthy control group (*p* < 0.05). The mean BMI of obese patients without fatty liver was considerably higher than that of the control group (*p* = 0.001). There was no statistically noteworthy variance between the obese patients without fatty liver and the control group in HbA1c levels (*p* > 0.05). 

Data on obese adolescents and healthy controls with and without fatty liver were presented in Table 2.

Correlations in the entire obese adolescent group (n:83): There was a positive and statistically important relationship between AIP and BMI, HbA1c, HOMA-IR, insulin, cholesterol/HDL cholesterol, and triglyceride/HDL cholesterol (*p* < 0.05). There were negative and statistically noteworthy correlations amongst AIP and cholesterol and AIP and cholesterol/triglyceride (*p* < 0.05).

Correlations in the total obese adolescent group were presented in Table 3.

Correlations in the obese adolescent group with only fatty liver (n:39): There was a positive and statistically significant relationship between AIP and BMI, HbA1c, HOMA-IR, insulin, cholesterol/ HDL cholesterol, and triglyceride/HDL cholesterol (*p* < 0.05). There was a negative and statistically substantial link between AIP and vitamin D levels and cholesterol/triglyceride ratio (*p* < 0.05).

Correlations in the obese adolescent group with fatty liver were presented in Table 4. 

A ROC curve was drawn for AIP in the diagnosis of fatty liver. The area under the curve is 0.757 and its standard error is 0.05. The area under the ROC curve was found to be significantly higher than 0.5 (*p* = 0.001). The cut-off point for AIP in the diagnosis of fatty liver is >0.41. The sensitivity of this value was 61.5% and the specificity was 83.5% (Area under the ROC curve (AUC) = 0.757; 95% cervical length (CI) = 0.676–0.826; *p* = 0.001). A ROC curve for AIP in the diagnosis of fatty liver was shown in Figure 1.

## 4. Discussion

NAFLD, which starts with the collection of fat in hepatocytes in the absence of extreme alcohol intake, is associated with obesity, type 2 diabetes, hyperlipidemia, and insulin resistance. Worldwide, the NAFLD incidence is increasing [29,30]. Generally, NAFLD is asymptomatic and is a silent disease that can be identified in basic health examinations with some biochemical changes in liver enzymes without any other particular cause, such as alcohol consumption, virus infection, drug effects, or autoimmune diseases. The diagnosis of NAFLD entails the validation of hepatic steatosis based on imaging studies or liver biopsies with clinical examinations [31,32]. AIP is calculated from laboratory parameters such as triglycerides and HDL cholesterol and, according to the literature, is associated with liver fattening.

Wang et al. divided obese people with NAFLD into three groups according to their AIP values and found that the prevalence of NAFLD increased in the medium- and high-risk groups. They found that AIP and total cholesterol, LDL cholesterol, and BMI levels were associated with NAFLD, but AIP, in particular, showed a much stronger association with NAFLD. They stated that AIP may be an indicator that contributes to the diagnosis of NAFLD [15]. Risal et al. revealed that those with grade 3 liver fat had higher AIP values than those with grade 1 and 2 liver fat. However, they could not find a difference between the total NAFLD group and the healthy control group in terms of AIP values. Since NAFLD is significantly associated with atherogenic dyslipidemia, they stated that AIP may be useful in evaluating dyslipidemia to prevent CVD [16]. Xie et al. reported higher BMI, ALT, AST, and AIP values in NAFLD patients. They stated that those with high AIP values were more likely to be male and younger and had higher BMI, ALT, and AST values. They found in their study that those with higher AIP levels tended to have a higher risk of NAFLD. They also stated that AIP showed the strongest relationship with fatty liver compared to other parameters; it was the strongest biomarker for NAFLD and could be a reference index in diagnosis and treatment [17]. Dong et al. stated that AIP and NAFLD showed a positive correlation in Chinese and Japanese people and could be a new screening sign for non-obese people with NAFLD in different countries. They stated that age, GGT, ALT, albumin, uric acid, glucose, LDL cholesterol, BMI, and AIP are independent factors that have a positive correlation with the progression of fatty liver disease, but AIP is the most notable independent factor among them [18]. Fadaei et al. reported high AIP values in NAFLD patients compared to healthy controls. They found that AIP had an independent relationship with carotid intima-media thickness in NAFLD patients [19].

In our study, we calculated the AIP rates in the obese adolescent group with and without NAFLD. We found that the mean AIP, BMI, HOMA-IR, and insulin levels of obese adolescents with fatty liver were higher than those of obese adolescents without fatty liver and the healthy control group. Again, the mean AIP of obese patients without fatty liver was significantly higher than that of the healthy control group. Therefore, those with the highest AIP values were those with fatty liver. Our findings are consistent with other studies.

However, studies on the relationship between fatty liver and AIP in pediatric patients are limited. The only study we found was by Ünal et al. [20] with 172 obese children and adolescents between 7 and 18 years of age. They reported a substantial positive connection between AIP and steatosis. They stated that AIP is a simple, inexpensive, and easily calculable parameter that can be a functioning indicator to predict NAFLD and CVD risk in obese children.

The definitive diagnosis of fatty liver is made by pathological examination. The discriminative power of the USG examination is limited, especially in grade 1 fatty liver. Therefore, we did not include those with USG grade 1 fatty liver. Furthermore, we did not make a separate classification for grades 2 and 3. Our study is the first in the literature to examine the rate of liver fattening and AIP in the adolescent group since the study sample consists of obese people between 10 and 17 years of age. According to our current study data and the results of other research, the existence of AIP and fatty liver are related. High AIP levels may be beneficial as a strong indicator, especially for grade 2 or 3 fatty liver disease (NAFLD).

NAFLD patients have a higher risk of atherosclerosis and CVD. In patients with NAFLD, AIP may increase, and higher AIP values may be associated with an increased risk of atherosclerosis and CVD. There is a significant increase in carotid intima-media thickness—an indicator of early atherosclerosis in patients with NAFLD [33,34]. Several studies on children have shown the relationship between early atherosclerosis, carotid intima-media thickness, and NAFLD [35]. In another study with a pediatric sample, carotid intima-media thickness was significantly higher in obese children with NAFLD than in healthy children or obese children without NAFLD [36]. Previous findings were in a large pediatric study evaluating the relationship between childhood NAFLD and CVD risk factors [37]. Autopsy findings of 817 children who passed away from external causes showed that children with liver fat had significantly more common atherosclerosis than those without liver fat [38].

One of the most critical known risk factors for CVD is dyslipidemia. The decrease in HDL cholesterol and increase in total cholesterol, LDL cholesterol, and triglyceride levels contribute to the progression of the atherosclerotic process [39]. Due to their small particle size, small-density LDL particles (sdLDL) can penetrate the arterial walls, accumulate for easy storage, and oxidize to oxidized LDL (OxLDL) compared to HDL particles. When OxLDL is phagocytized by macrophages, macrophages turn into foam cells, which cause atherosclerosis and CVD. Studies suggest the clinical use of sdLDL particles as an indicator for predicting atherosclerosis [40]. As noted in the Dobiasova and Frohlich studies, the AIP value is inversely proportional to the circumference of LDL particles and reflects the size of sdLDL particles. Therefore, this measure reflects the balance between protective and atherogenic lipoproteins [41]. Lipid ratios such as total cholesterol/HDL cholesterol and LDL cholesterol/HDL cholesterol are determinants of CVD [39,41]. However, studies have reported that the AIP value is a more consistent indicator for CVD risk factors than traditional lipid parameters and lipid ratios [42,43]. In this respect, high AIP values in adolescents with liver fattening in our study may be predictive for atherosclerotic cardiovascular diseases. In clinical practice, an accurate index is necessary to screen for and predict NAFLD. Reporting the AIP value in laboratory results can also be cautionary and noteworthy for clinicians. In addition to NAFLD, this application may be beneficial in predicting atherosclerosis and CVD in treatment follow-up. More studies are necessary on the subject, especially in pediatric cases.

In this study, we found a positive correlation between AIP values and cholesterol/HDL cholesterol and triglyceride/HDL cholesterol ratios in both the total obese group and the obese group with fatty liver. Additionally in these groups, there was a negative and statistically important connection between AIP and cholesterol/triglyceride ratio. We found that the cholesterol/HDL cholesterol and triglyceride/HDL cholesterol ratios were higher in the obese group. We found that this increase was higher in obese patients with fatty liver. Our findings demonstrate the association of all lipid levels with fatty liver and obesity. However, based on literature data [15,16,17,18,41,42,43] and our study’s findings, we think that it is more valuable to calculate AIP levels and show their relationship with fatty liver. In addition, many studies continue to show the superiority of AIP. Our study will also contribute to other lipid ratio studies.

Vitamin D_3_ plays a role in many biological processes, such as the regulation of calcium and phosphorus metabolism and the prevention of cardiovascular disease and inflammation. Vitamin D_3_ deficiency may cause various metabolic diseases [44,45,46,47,48], NAFLD [49], and a decrease in bone mineral content. Hypovitaminosis D is associated with the severity [49] and incidence of NAFLD in patients with normal liver enzymes [50]. According to the results of a meta-analysis, 26% of patients with NAFLD had a vitamin D deficiency [51].

We did not find any difference in vitamin D levels between our patient groups. Indeed, it was low in all groups. In addition, in this research, we found a negative, reasonable relationship between AIP and vitamin D only in the group with fatty liver. Our findings are consistent with the studies in the literature [26,27]. However, interestingly, we could not find a correlation between AIP and vitamin D in the entire obese group, while we only found a correlation between AIP and vitamin D in the group with fatty liver. İzadi et al. found that serum vitamin D levels were negatively associated with AIP in NAFLD patients [52]. The literature supports the inverse relationship between AIP and vitamin D in our study. In another study, Pokhrel et al. observed a significant negative correlation between vitamin D with lipid markers and atherogenic variables in a poor glycemic control diabetic population [53]. Some studies have shown that vitamin D supplementation improves dyslipidemia and atherogenic indices [54], and may be effective in preventing atherosclerosis [55]. Therefore, keeping vitamin D at optimal levels may prevent fatty liver if we supplement deficient children. Additionally, we believe that keeping AIP values low together with vitamin D at optimal levels may have a preventive effect on atherosclerosis and CVD in those with fatty liver.

The presence of NAFLD was not determined by liver biopsy; fatty liver was not graded due to the low distinguishing power of USG, and the relatively low number of patients are some of the limitations of our study. Moreover, most adolescents with fatty were male. However, we think that our study will guide other studies on this subject.

## 5. Conclusions

This study found that AIP levels increased in obese adolescents, and this increase was higher in obese patients with fatty liver. In addition, we detected a negative correlation between AIP and vitamin D levels, and a positive correlation with BMI, insulin resistance, and insulin levels. Based on our data, we concluded that AIP can be used for predicting fatty liver in obese adolescents.

## Figures and Tables

**Figure 1 children-10-00641-f001:**
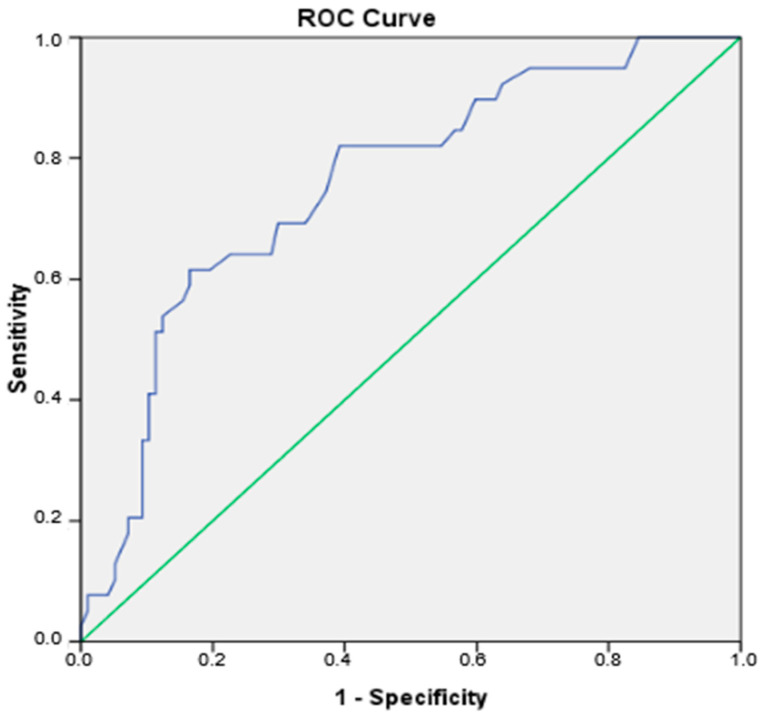
A ROC curve for AIP in the diagnosis of fatty liver.

**Table 1 children-10-00641-t001:** Data on obese adolescents and healthy controls.

	Obese Adolescents (n = 83)	Healthy Control (n = 53)	
	Mean ± SD	Mean ± SD	*p*-Value
Serum Biochemical Parameters:			
HbA1c (%)	5.42 ± 0.37	5.33 ± 0.31	0.161 ^1^
Glucose (mg/dL)	89.73 ± 8.03	89.36 ± 7.5	0.785 ^1^
Urea (mg/dL)	23.49 ± 6.24	22.38 ± 5.85	0.299 ^1^
Creatinine (mg/dL)	0.57 ± 0.14	0.58 ± 0.15	0.911 ^1^
AST (U/L)	25.2 ± 12.86	19.68 ± 5.17	0.001 ^2^
ALT (U/L)	28.78 ± 25.38	14.21 ± 7.06	0.001 ^2^
CRP (mg/L)	4.89 ± 4.47	3.13 ± 3.51	0.003 ^2^
Cholesterol (mg/L)	162.33 ± 33.53	146.96 ± 28.98	0.007 ^1^
Triglyceride (mg/L)	123.2 ± 60.67	83.72 ± 34.15	0.001 ^2^
HDL cholesterol (mg/dL)	44.27 ± 9.14	51.66 ± 11.3	0.001 ^1^
LDL cholesterol (mg/dL)	98.45 ± 29.69	80.96 ± 26.13	0.001 ^1^
TSH (mU/L)	2.78 ± 1.3	2.66 ± 1.76	0.641 ^1^
Free T4 (ng/L)	10.31 ± 2.23	10.3 ± 2.21	0.991 ^1^
25-Hydroxyvitamin D_3_ (ug/L)	16.91 ± 8.96	16.58 ± 6.45	0.853 ^2^
Insulin (IU/L)	22.68 ± 15.37	10.66 ± 5.38	0.001 ^2^
Calculated indexs:			
HOMA-IR	5.07 ± 3.61	2.37 ± 1.25	0.001 ^2^
Cholesterol/HDL cholesterol	3.81 ± 1.14	2.96 ± 0.84	0.001 ^2^
Triglyceride/HDL cholesterol	3.02 ± 1.89	1.71 ± 0.79	0.001 ^2^
AIP	0.40 ± 0.26	0.19 ± 0.20	0.001 ^1^
Cholesterol/triglyceride	1.58 ± 0.69	1.97 ± 0.73	0.002 ^2^

^1^ Student’s *t*-test; ^2^ Mann–Whitney U Test. AIP: Atherogenic index of plasma, ALT: Alanine aminotransferase, AST: Aspartate aminotransferase, BMI: Body mass index, CRP: C-reactive protein, HOMA-IR: Homeostatic model of assessment for insulin resistance, HDL cholesterol: High-density lipoprotein cholesterol, LDL cholesterol: Low-density lipoprotein cholesterol, SD: Standard deviation, TSH: Thyroid-stimulating hormone.

**Table 2 children-10-00641-t002:** Data on obese adolescents and healthy controls with and without fatty liver.

	Obesity (+) Fatty Liver (−) (n = 44)	Obesity (+) Fatty Liver (+) (n = 39)	Healthy Control (n = 53)	
	Mean ± SD	Mean ± SD	Mean ± SD	*p*-Value
Demographic variables:				
Age	13.02 ± 2.01	12.79 ± 1.95	13.3 ± 1.9	0.462 ^1^
Sex n.%				
Female	28 (63.6%)	7 (17.9%)	27 (50.9%)	0.001 ^3^
Male	16 (36.4%)	32 (82.1%)	26 (49.1%)	
BMI (kg/m^2^)	30.02 ± 3.45 ^b^	32.26 ± 5.39 ^a,b^	21.62 ± 3.03	0.001 ^1^
Serum Biochemical Parameters:				
HbA1c (%)	5.31 ± 0.33	5.54 ± 0.37 ^a,b^	5.33 ± 0.31	0.003 ^1^
Glucose (mg/dL)	87.86 ± 7.95	91.85 ± 7.68	89.36 ± 7.5	0.064 ^1^
Urea (mg/dL)	22.73 ± 6.41	24.36 ± 6	22.38 ± 5.85	0.279 ^1^
Creatinine (mg/dL)	0.56 ± 0.1	0.59 ± 0.18	0.58 ± 0.15	0.635 ^1^
AST (U/L)	21.18 ± 6.76	29.74 ± 16.29 ^a,b^	19.68 ± 5.17	0.001 ^2^
ALT (U/L)	18.55 ± 7.74 ^b^	40.33 ± 32.62 ^a,b^	14.21 ± 7.06	0.001 ^2^
CRP (mg/L)	4.5 ± 4.61 ^b^	5.33 ± 4.31 ^b^	3.13 ± 3.51	0.006 ^2^
Cholesterol (mg/L)	161.36 ± 35.61	163.41 ± 31.44 ^b^	146.96 ± 28.98	0.025 ^1^
Triglyceride (mg/L)	111.84 ± 58.28 ^b^	136.03 ± 61.5 ^a,b^	83.72 ± 34.15	0.001 ^2^
HDL cholesterol (mg/dL)	46.89 ± 8.97 ^b^	41.31 ± 8.5 ^a,b^	51.66 ± 11.3	0.001 ^1^
LDL cholesterol (mg/dL)	98.68 ± 31.63 ^b^	98.18 ± 27.75 ^b^	80.96 ± 26.13	0.003 ^1^
TSH (mU/L)	2.66 ± 1.25	2.91 ± 1.36	2.66 ± 1.76	0.680 ^1^
Free T4 (ng/L)	10.23 ± 2.26	10.4 ± 2.22	10.3 ± 2.21	0.943 ^1^
25-Hydroxyvitamin D_3_ (ug/L)	18.23 ± 10.52	15.42 ± 6.62	16.58 ± 6.45	0.573 ^1^
Insulin (IU/L)	17.9 ± 10.11 ^b^	28.07 ± 18.39 ^a,b^	10.66 ± 5.38	0.001 ^2^
Calculated indexs:				
HOMA-IR	3.88 ± 2.29 ^b^	6.4 ± 4.33 ^a,b^	2.37 ± 1.25	0.001 ^2^
Cholesterol/HDL cholesterol	3.58 ± 1.18 ^b^	4.08 ± 1.05 ^a,b^	2.96 ± 0.84	0.001 ^2^
Triglyceride/HDL cholesterol	2.61 ± 1.78 ^b^	3.49 ± 1.92 ^a,b^	1.71 ± 0.79	0.001 ^2^
AIP	0.33 ± 0.26 ^b^	0.48 ± 0.23 ^a,b^	0.19 ± 0.2	0.001 ^1^
Cholesterol/triglyceride	1.73 ± 0.71	1.42 ± 0.62 ^a,b^	1.97 ± 0.73	0.001 ^2^

^1^ One-way ANOVA test. ^2^ Kruskal-Wallis Test. ^3^ Chi-square test. ^a^ Versus to obesity (+) fatty liver (−) group ^b^ Versus to control group. AIP: Atherogenic index of plasma, ALT: Alanine aminotransferase, AST: Aspartate aminotransferase, BMI: Body mass index, CRP: C-reactive protein, HOMA-IR: Homeostatic model of assessment for insulin resistance, HDL cholesterol: High-density lipoprotein cholesterol, LDL cholesterol: Low-density lipoprotein cholesterol, SD: Standard deviation, TSH: Thyroid-stimulating hormone.

**Table 3 children-10-00641-t003:** Correlations in the total obese adolescent group (n:83).

Total Obese		Cholesterol/HDL	Triglyceride/HDL	Cholesterol/Triglyceride	AIP
BMI	r	0.247	0.370	−0.204	0.304
	*p*-value	0.024	0.001	0.064	0.005
HbA1c	r	0.281	0.276	−0.142	0.269
	*p*-value	-	0.012	-	0.014
Cholesterol	r	-	0.374	-	−0.355
	*p*-value	0.001	0.001	0.695	0.001
Triglyceride	r	0.651	-	-	-
	*p*-value	^+^ 0.001	^-^	^-^	-
HDL cholesterol	r	-	-	0.513	-
	*p*-value	-	-	0.001	-
LDL cholesterol	r	0.605	0.215	0.111	0.188
	*p*-value	0.001	0.051	0.318	0.089
25-Hydroxyvitamin D_3_	r	−0.040	−0.096	0.140	−0.206
	*p*-value	^+^ 0.719	^+^ 0.386	^+^ 0.207	0.062
HOMA-IR	r	0.245	0.398	0.403	0.322
	*p*-value	0.026	^+^ 0.001	^+^ 0.001	0.003
Insulin	r	0.268	0.401	−0.397	0.325
	*p*-value	0.014	^+^ 0.001	^+^ 0.001	0.003
Cholesterol/HDL cholesterol	r	-	-	-	0.734
	*p*-value	-	-	-	0.001
Triglyceride/HDL cholesterol	r	-	-	-	0.946
	*p*-value	-	-	-	0.001
Cholesterol/triglyceride	r	-	-	-	−0.879
	*p*-value	-	-	-	0.001

Pearson Correlation Analysis. ^+^ Spearman’s rho correlation analysis.

**Table 4 children-10-00641-t004:** Correlations in the obese adolescent group with fatty liver (n:39).

Fatty liver (+)		Cholesterol/HDL	Triglyceride/HDL	Cholesterol/Triglyceride	AIP
BMI	r	0.380	0.454	−0.253	0.375
	*p*-value	0.017	0.004	0.120	0.019
HbA1c	r	0.265	0.184	−0.007	0.133
	*p*-value	0.103	0.263	0.967	0.419
	*p*-value	^+^ 0.404	^+^ 0.953	^+^ 0.714	0.989
Cholesterol	r	-	0.319	-	0.277
	*p*-value	-	0.048	-	0.088
Triglyceride	r	0.412	-	-	-
	*p*-value	^+^ 0.009	^-^	^-^	-
HDL cholesterol	r	-	-	0.326	-
	*p*-value	-	-	0.043	-
LDL cholesterol	r	0.585	0.150	0.203	0.116
	*p*-value	0.001	0.361	0.215	0.483
25-Hydroxyvitamin D_3_	r	−0.343	−0.331	0.256	−0.373
	p-value	^+^ 0.032	0.040	^+^ 0.116	0.019
HOMA-IR	r	0.304	0.587	–0.500	0.449
	*p*-value	^+^ 0.060	^+^ 0.001	^+^ 0.001	0.004
Insulin	r	0.313	0.595	−0.510	0.467
	*p*-value	^+^ 0.052	^+^ 0.001	^+^ 0.001	0.003
Cholesterol/HDL cholesterol	r	-	-	-	0.597
	*p*-value	-	-	-	0.001
Triglyceride/HDL cholesterol	r	-	-	-	0.825
	*p*-value	-	-	-	0.001
Cholesterol/triglyceride	r	-	-	-	−0.807
	*p*-value	-	-	-	0.001

Pearson Correlation Analysis. ^+^ Spearman’s rho correlation analysis.

## Data Availability

Data is contained within this article [Table 1, Table 2 and Table 3].

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
