# Peer review of "Atherogenic Index of Plasma (AIP) and Its Association with Fatty Liver in Obese Adolescents"

_children, 2023, doi:10.3390/children10040641_

Round 1

Reviewer 1 Report

This study aimed to show that AIP, a calculated indicator for atherogenic disorders, correlated with fatty liver in adolescent. Although the results can be published, several parts of the manuscript could be improved/added to. Here are some examples.

- Was the sample size calculation performed?  Was n sufficient for statistical calculation? 

- When and where the study performed? What are the ethic group of those adolescents? Who did USG? Who diagnosed grading of fatty liver? 

- What is the cut-off value of AIP to indicate disease stage or severity?

- Line 140-142 and in the abstract, if the authors provided the exact p-value, p<0.05 is not necessary.

- Table 1 requires extensive improvement, both on formatting and arrangement. The row could be divided into demographic info, serum biochemical parameters, and calculated index. 

- It would be better do display the parameters of overall obese adolescent before dividend into +/-fatty liver in table 1.

- I don't understand why the author use "median" instead of "mean" for most of parameters in table 1? Are there any reasons?

- What is the aim to show those p-value in table 1? To show that there was a different among 3 groups, or to show that one group is significantly different from another group, so what is that group. According to the study objective, please critically consider what should be shown.

- Again, the signal * is not necessary if the exact p-values are provided. It should preferably be used to indicate types of statistics. It's ambiguous currently.

- Most of fatty liver adolescents was male. Is it affect the finding? Is this a selection bias of this study?

- In table 2 and 3, it does not make sense to show the correlation between the same set of parameters; for example, cholesterol vs cholesterol/HDL, since cholesterol/HDL is calculated using cholesterol. So the information of these tables should be extensively revised and merged if possible.

- It would be better to show the correlation between calculated parameters such as the authors did for HOMA-IR vs AIP. How is cholesterol/HDL vs AIP, and so on?

- It seems like the results on vitamin D was shown by an accident, since there is only a parameter. I would suggest considering to renew the study title.

- Since this study aimed to show the utility of AIP, the pro- and con of AIP when compared to other parameters should be extensively discussed. Now, most of the discussion part it not necessary for the readers.

- Due to API is calculated using triglyceride and HDL, why don't just use the ratio between triglycerides/HDL, as shown in the result part?

- The conclusion is ambiguous. I think the authors are not capable to recommend people what to do, but the authors may provide that AIP can be used for predicting NAFLD and so on.

- What is the stool vitamin D?

- The phase "Based on our data, we concluded that low AIP levels in the adolescent age group and keeping vitamin D stools at optimal levels are critical for liver fattening and the protection of cardiovascular diseases." is unclear. 

Author Response

Dear Editor and Dear Reviewer;

First of all thank you very much for your efforts and your comments on our manuscript. We have changed our study based on the reviewer's comments. The file is attached.

Reviewer 2 Report

Althought most of paper are about adolts

Please add these references

Associations between serum vitamin D3, atherogenic indices of plasma and cardiometabolic biomarkers among patients with diabetes in the KERCADR study. Mahmoodi MR, Najafipour H.Mahmoodi MR, et al. BMC Endocr Disord. 2022 May 12;22(1):126. doi: 10.1186/s12902-022-01043-1.BMC Endocr Disord. 2022. PMID: 35549686 Free PMC article. Clinical Trial. Independent t-test, Hierarchical Linear Regression, Univariate ANOVA, and partial correlation were used for analysis the data. Atherogenic indices of plasma include Castelli Risk Index I (CRI I), Castelli Risk Index II (CRI II), and the novel Athero … Independent t-test, Hierarchical Linear Regression, Univariate ANOVA, and partial correlation were used for analysis the data. Atherogeni … Cite Share Item in Clipboard 2 Cite Share Atherogenic Index of Plasma and Its Association with Risk Factors of Coronary Artery Disease and Nutrient Intake in Korean Adult Men: The 2013-2014 KNHANES. Shin HR, Song S, Cho JA, Ly SY.Shin HR, et al. Nutrients. 2022 Mar 3;14(5):1071. doi: 10.3390/nu14051071.Nutrients. 2022. PMID: 35268046 Free PMC article. Coronary artery disease (CAD) has been linked to one of the highest death rates globally. The atherogenic index of plasma (AIP) may be an important predictor of atherosclerosis and cardiovascular disease, superior to the standard atherosclerotic lipid profile … Coronary artery disease (CAD) has been linked to one of the highest death rates globally. The atherogenic index of plasma … Cite Share Item in Clipboard 3 Cite Share Vitamin D Replacement Mitigates Menopause-Associated Dyslipidaemia and Atherogenic Indices in Ovariectomized Rats; A Biochemical Study. Muhammad MH, Hussien NI, Elwia SK.Muhammad MH, et al. Exp Clin Endocrinol Diabetes. 2020 Mar;128(3):144-151. doi: 10.1055/a-0934-5666. Epub 2019 Jun 24.Exp Clin Endocrinol Diabetes. 2020. PMID: 31234220 So, it becomes prudent to search for natural safe alternatives. Vitamin D (VD) has been acknowledged as an essential factor in cardiovascular health. ...RESULTS: Dyslipidaemia with an increased atherogenic index of plasma, atherosclerosis coeffi … So, it becomes prudent to search for natural safe alternatives. Vitamin D (VD) has been acknowledged as an essential factor in … Cite Share Item in Clipboard 4 Cite Share Involvement of RBP4 in Diabetic Atherosclerosis and the Role of Vitamin D Intervention. Zhou W, Ye SD, Chen C, Wang W.Zhou W, et al. J Diabetes Res. 2018 Aug 16;2018:7329861. doi: 10.1155/2018/7329861. eCollection 2018.J Diabetes Res. 2018. PMID: 30186876 Free PMC article. The purposes of this study were to evaluate the expression of retinol-binding protein 4 (RBP4) in diabetic rats with atherosclerosis and to investigate the role of vitamin D intervention. Male Wistar rats were randomly divided into 4 groups, including the control gr … The purposes of this study were to evaluate the expression of retinol-binding protein 4 (RBP4) in diabetic rats with atherosclerosis and to … Cite Share Item in Clipboard 5 Cite Share Vitamin D insufficiency and its association with adipokines and atherogenic indices in patients with metabolic syndrome: A case-control study. Amirkhizi F, Khademi Z, Hamedi Shahraki S, Rahimlou M.Amirkhizi F, et al. Front Endocrinol (Lausanne). 2023 Jan 20;14:1080138. doi: 10.3389/fendo.2023.1080138. eCollection 2023.Front Endocrinol (Lausanne). 2023. PMID: 36742396 Free PMC article. INTRODUCTION: Vitamin D deficiency is one of the most common nutritional disorders in most countries of the world. ...Atherogenic index of plasma (AIP) was calculated as log (TG/HDL-c). ... INTRODUCTION: Vitamin D deficiency is one of the most common nutritional disorders in most countries of the world. ...Ather … Cite Share Item in Clipboard 6 Cite Share Plasma vitamin D and parathormone are associated with obesity and atherogenic dyslipidemia: a cross-sectional study. Guasch A, Bulló M, Rabassa A, Bonada A, Del Castillo D, Sabench F, Salas-Salvadó J.Guasch A, et al. Cardiovasc Diabetol. 2012 Dec 11;11:149. doi: 10.1186/1475-2840-11-149.Cardiovasc Diabetol. 2012. PMID: 23228198 Free PMC article. BACKGROUND: Low concentrations of plasma vitamin D (25(OH)D) have been associated with the development of metabolic syndrome (MetS), obesity, diabetes and cardiovascular disease. The objective of this study was to quantify the associations between 25(O … BACKGROUND: Low concentrations of plasma vitamin D (25(OH)D) have been associated with the development of metabo … Cite Share Item in Clipboard 7 Cite Share Vitamin D deficiency and cardiovascular risk in type 2 diabetes population. Pokhrel S, Giri N, Pokhrel R, Pardhe BD, Lamichhane A, Chaudhary A, Bhatt MP.Pokhrel S, et al. Open Life Sci. 2021 May 10;16(1):464-474. doi: 10.1515/biol-2021-0050. eCollection 2021.Open Life Sci. 2021. PMID: 34017921 Free PMC article. Cardiac risk ratio, atherogenic index plasma, and atherogenic coefficient were calculated to assess and compare the CVD risk in different groups. ...We also observed significant negative correlation of vitamin D with lipid markers and … 

Author Response

First of all thank you very much for your efforts and your comments on our manuscript. We have changed our study based on the reviewer's comments. The file is attached.

Round 2

Reviewer 1 Report

Despite some improvement, I would again provide/repeat suggestions for a better manuscript.

- For sample size, I don't want to know. However, it is a must to include and declare to the readers. Then the reader would consider if your study were good enough or not. Such as in your case, if the authors stated that 17/group was the minimum, why the N is > 39.

- For the AIP cut-off value, since there was the association between 2 parameters and the authors also suggest that AIP could be used to predict fatty liver, the authors should be capable to suggest from the study results. It would add the value to the publication. Otherwise, this conclusion is untrue.

- HOMA-IR is a calculated parameter.

- The caption under each table is needed to be rewrite. For an example on table 1, the authors have to find a way to indicate what student t-test or Mann-Whitney test were applied to each row.

- The 3rd column of table 2 is just a repeat of table 1, not necessary. Again, what is the p-value in table 2, what is the comparison.

- Comma and dot are madly used in every table.

- Since there is no supporting info, still, I would suggest not to mention about vitamin D on the title. It's acceptable in the result and discussion.

Author Response

Dear Editor and Dear Reviewer;

First of all thank you very much for your efforts and your comments on our manuscript.

We have changed our study based on the reviewer comments.
